# *In vivo* study of the radioadaptive response and low-dose hyper-radiosensitivity for chromosome breaks induced by gamma rays in wild-type *Drosophila melanogaster* larval neuroblasts: Dose and dose rate dependence

Claudia Di Dio[1,2☉], Antonella Porrazzo[2☉], Alex De Gregorio[3], Patrizia Morciano[4], Maria Antonella Tabocchini[5], Giovanni Cenci[2,6], Francesca Cipressa[7*], Giuseppe Esposito[1,5*]

1 Istituto Superiore di Sanità (ISS), Rome, Italy, 2 Dipartimento di Biologia e Biotecnologie "C. Darwin", Sapienza Università di Roma, Rome, Italy, 3 Laboratorio SAFU, Dipartimento di Ricerca, Diagnostica Avanzata e Innovazione Tecnologica, Area Ricerca Traslazionale, IRCCS Istituto Nazionale Tumori Regina Elena, Rome, Italy, 4 Dipartimento di Medicina Clinica, Sanità Pubblica, Scienze della Vita e dell'Ambiente, Università Degli Studi dell'Aquila, L'Aquila, Italy and Laboratori Nazionali del Gran Sasso (LNGS), INFN, Assergi, L'Aquila, Italy, 5 INFN-Roma 1, Rome, Italy, 6 Istituto Pasteur Italia-Fondazione Cenci Bolognetti, Rome, Italy, 7 Dipartimento di Scienze Biologiche ed Ecologiche, Università degli Studi della Tuscia, Viterbo, Italy

☉ These authors contributed equally to this work.
* giuseppe.esposito@iss.it

## Abstract

Although the biological effects of low doses/dose rates of ionising radiation have been extensively studied both *in vitro* and *in vivo*, there are still knowledge gaps to be filled. For example, the mechanisms underlying the phenomena of radioadaptive responses and hypersensitivity to low doses of radiation are still not fully understood. This study aims to investigate the phenomenon of radioadaptive response in *Drosophila melanogaster* larval neuroblasts, focusing on the influence of different gamma priming doses and priming dose rates. We examined the modulation of cytogenetic damage, specifically the frequency of chromosome breaks, induced by a challenging dose of 10 Gy following different priming doses (0–2.7 Gy) delivered at dose rates ranging from 1.4 to 17 mGy/h. Our findings reveal the presence of a distinct window in which radioadaptive responses occurs, notably above a certain threshold dose when delivered at a rate of 1.4 mGy/h. Consistently with our previous results, we confirmed that the maximal protection was observed at a priming dose of 0.4 Gy delivered at 2.5 mGy/h. Additionally, we studied the occurrence of chromosome breaks after irradiating larval neuroblasts at doses ranging from 0.7 to 10 Gy. Notably, in this case we observed a low-dose hyper-radiosensitivity phenomenon up to 2.7 Gy, followed by increased resistance above 2.7 Gy. Our results provide insight into the complex cellular responses to low-dose/dose rate radiation and have implications in various fields, including radiation protection, diagnostics, theragnostics and biodosimetry.

**Data availability statement:** All relevant data are within the manuscript and its Supporting Information files.

**Funding:** Financial support was granted by Istituto Superiore di Sanità (Italian National Institutes of Health). The funders had no role in study design, data collection and analysis, decision to publish, or preparation of the manuscript.

**Competing interests:** The authors have declared that no competing interests exist.

## Introduction

It has been amply demonstrated that a biological system exposed to high acute doses of ionising radiation suffers severe damage that can potentially lead to its death. However, biological effects of low-dose (below 100 mGy) or low-dose rate (below 5 mGy/h) IR remain elusive [1]. In general, low doses primarily exert a modulatory effect on normal metabolism. Furthermore, at low doses, a spatial heterogeneity in the energy released by radiation on the biological system is often obtained. Probably the major actor solicited by low dose irradiations is the cellular communication, which probably contributes to low-dose phenomena such as, bystander effects, adaptive responses, low-dose hypersensitivity and genomic instability [2]. Some of these effects, particularly adaptive response and low-dose hypersensitivity, may share common mechanisms.

The response of cells, tissues and organisms to a given acute radiation exposure can often be modified if a small conditioning dose (called the "priming dose") is delivered a few hours before this acute radiation insult (called "challenging dose). This effect is referred as to radioadaptive response (RAR). Most of the experiments that have investigated the phenomenon of RAR have been carried out using *in vitro* biological systems and low linear energy transfer (LET) radiations. The earliest evidence of RAR was reported in human lymphocytes [3], and subsequent studies have demonstrated RAR in various eukaryotic cells using different indicators of cellular damage, including cell lethality, transformation, chromosomal aberrations, micronuclei formation, gene mutations, and DNA double-strand break repair (see, for example, [4] for a review). Although *in vivo* studies on RAR are less common than *in vitro* studies, they have been conducted using models such as mice, rabbits, and Leopard frogs. In contrast to *in vitro* studies where priming doses are delivered over timescales of minutes or a few hours, in *in vivo* studies priming doses are usually protracted over timescales of days or weeks [5]. *Drosophila melanogaster* has also been used as an *in vivo* biological system to investigate the phenomenon of RAR. Moskalev et al. found that *FOXO*, *SIRT1*, *JNK*, *ATM*, *ATR*, and *p53* genes play an essential role in RAR of the whole organism on *Drosophila melanogaster* lifespan (using a chronic priming dose of 0.4 Gy at a dose rate of 1.7 mGy/h) [6]. The results obtained by Koval et al showed that the activity of DNA repair genes is essential for the RAR and hormesis on *Drosophila melanogaster* lifespan (using a chronic priming dose of 0.4 Gy at a dose rate of 1.4 mGy/h). Apparently, radioprotective effects of DNA repair genes are associated with activation of different mechanisms of cell resistance to stress [7]. In a previous study we found that in wild-type *Drosophila melanogaster* larval neuroblasts, the frequency of chromosome breaks (CBs), induced by acute gamma irradiation, was considerably reduced when flies were previously exposed to a protracted gamma dose of 0.4 Gy delivered at a dose rate of 2.5 mGy/h. Deep RNA sequencing revealed that RAR was associated with the downregulation of *Loquacious D* (*Loqs-RD*) gene that encodes a well-conserved dsRNA binding protein required for esiRNAs biogenesis identifying Loqs as a key factor in low-dose radioresistance [8].

A seemingly distinct phenomenon from RAR is low-dose hyper-radiosensitivity (HRS), an effect in which biological systems exhibit unusually high radiosensitivity to low radiation doses [9–14]. As the dose increases, a transition to increased radioresistance occurs, where cells become more resistant per unit dose, a phenomenon that was named increased radioresistance (IRR).

In the present study, we investigated both RAR and HRS/IRR in *Drosophila melanogaster* larval neuroblasts by performing *in vivo* experiments. For the first time, the dependence of the RAR on priming dose and priming dose rate values of gamma rays in wild-type *Drosophila melanogaster* larval neuroblasts was examined. Additionally, for the first time, we analyzed the dose-response relationship for chromosome breaks in larval neuroblasts from third-instar *Drosophila* larvae to assess HRS/IRR.

## Methods

### *Drosophila* strain

The *Oregon* R strain was used in all experiments. Flies were maintained on *Drosophila* medium (Nutri-Fly®GF; Genesee Scientific) treated with propionic acid used as an antifungal.

### Irradiation treatments

All the irradiations were carried out at the Istituto Superiore di Sanità (ISS, Rome, Italy). For protracted treatments with gamma rays, the LIBIS irradiation facility housing a $^{137}$Cs source with activity of 19.4 GBq (as of August 24, 2012) was used [15]. For acute exposures a $^{137}$Cs gamma irradiator (Gammacell Exactor 40, Nordion Inc. Ottawa, Canada) was used. Vials containing 12 h *Drosophila* embryos (from 20 young females mated to 20 young males) were placed inside LIBIS (so as not to overlap) and irradiated continuously at dose rates of 1.4, 2.5, 4.4, 7.8 and 17 mGy/h until embryos developed into third instar larvae (7 days). During the entire exposure, the temperature was maintained at 23 °C. To obtain these dose rates, the vials were placed simultaneously inside the LIBIS irradiator at five different distances from the $^{137}$Cs source, namely d1 = 100 cm (for 1.4 mGy/h), d2 = 74 cm (for 2.5 mGy/h), d3 = 56 cm (for 4.4 mGy/h), d4 = 42 cm (for 7.8 mGy/h) and d5 = 28 cm (for 17 mGy/h) (Fig 1 and S1 Fig).

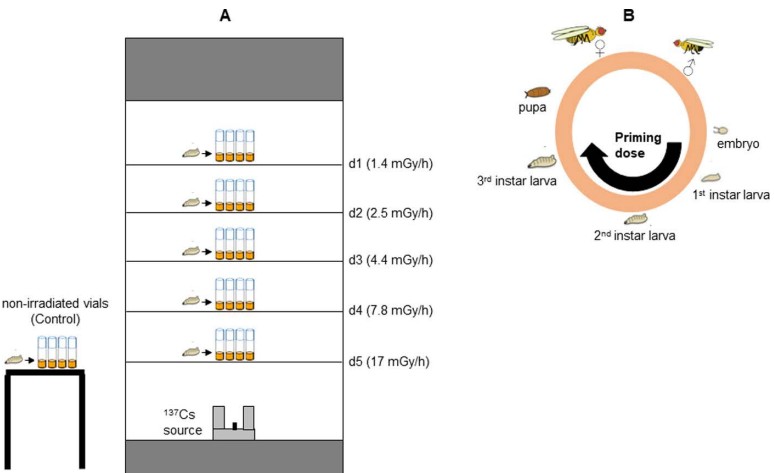

**Fig 1. Experimental plan.** (A) Schematic representation of the LIBIS irradiator. The vials were placed at five different distances (d1, d2, d3, d4 and d5) from the $^{137}$Cs source. Non-irradiated control vials were placed in the same room as the LIBIS irradiator at 23 °C. (B) The priming doses was delivered during the embryos-larvae transition. The third instar larvae were collected and exposed to the challenging dose.

The third instar larvae populated the vials from about the fourth day (89 h) up to about the seventh day (161 h) after the start of chronic irradiation. To select third instar larvae originating from embryos that received, throughout their development, a total priming dose of 0.1 Gy (at 1.4 mGy/h), 0.2 Gy (at 2.5 mGy/h), 0.4 Gy (at 4.4 mGy/h), 0.7 Gy (at 7.8 mGy/h) or 1.5 Gy (at 17 mGy/h), vials were irradiated for 89 h at the distance of d1, d2, d3, d4 and d5 respectively. We have also collected third instar larvae coming out after 161 h of maintenance inside LIBIS at d1, d2, d3, d4 and d5 distances that resulted into a total priming dose of 0.2 Gy (at 1.4 mGy/h), 0.4 Gy (at 2.5 mGy/h), 0.7 Gy (at 4.4 mGy/h), 1.3 Gy (at 7.8 mGy/h) and 2.7 Gy (at 17 mGy/h), respectively. The selected dose (0.4 Gy) and dose rate (2.5 mGy/h) values were based on previous findings demonstrating clear radioadaptive response [8]. This experimental design allows us to investigate whether our biological system exhibits adaptation within a specific range of priming dose and dose rate values.

Control non-pretreated vials were kept in parallel outside the LIBIS at 23 °C (Fig 1). Irradiated and control vials were exposed, 4 h after the end of the priming dose, to an acute challenging gamma radiation dose of 10 Gy at a dose rate of 0.65 Gy/min (values already used in the previous study [8]). Following challenging irradiation, non-irradiated and irradiated larvae were maintained in the same incubator at 25 °C for four hours and then they were dissected for the analysis of CBs frequency in neuroblasts. In addition, in order to obtain the dose-response curve for CBs induced by acute gamma rays, untreated third instar larvae were irradiated with doses of 0.7, 1.5, 2.7, 4, 6, 8 and 10 Gy at a dose rate of 0.65 Gy/min. Also in this case, following acute irradiation, the same procedure as described above was followed.

### *Drosophila* chromosome cytology and microscopy

To obtain metaphase chromosome preparations from *Drosophila* larval neuroblasts for the analysis of chromosome aberrations in metaphase mitotic chromosomes, third instar larval brains were dissected in a drop of 0.7% NaCl. The isolated brains were then transferred to a Petri dish containing $10^{-6}$ M colchicine. Samples were incubated for 1 hour at room temperature. Following incubation, brains were fixed in 0.5% sodium citrate hypotonic solution for 5 minutes and then they were transferred to a small drop of 45% acetic acid on coverslip and immediately frozen in liquid nitrogen. After removal of coverslips, slides were air dried and stained with DAPI/VECTASHIELD® (VECTOR Laboratories). For each condition, at least 100 metaphases were analyzed through direct observation using the inverted fluorescence microscope Nikon TE 2000 (Nikon Instruments Inc., Americas) equipped with a Charged-Coupled Device (CCD camera; Photometrics CoolSnap HQ).

### Statistical analysis

For each priming dose and priming dose rate value, at least three independent experiments were carried out. For each of these experiments the cell distribution of CBs, the average number of CBs per cell and the standard deviation of the cell distribution of CBs were determined (>100 cells were scored for each condition). The average of the mean values of CBs per cell obtained from at least three independent experiments was calculated together with its standard error (SE) for each condition. To determine statistical significant differences between pairs of these averages, the Graphpad software was used by performing a Dunnett test after a one-way ANOVA test. Values of of $p < 0.05$ were considered as statistically significant.

## Results

### RAR experiments

We investigated cytogenetic damage in *Drosophila* larval neuroblasts induced by irradiating third instar larvae with an acute challenging dose of 10 Gy preceded or not by pretreatment with different priming doses and different priming dose rates. We chose to consider only female larval neuroblasts, since females showed a higher frequency of both chromosomal and chromatid aberrations [8,16] and are generally more vulnerable to DNA damage and mutations [17]. Chromosomes were fixed four hours after acute irradiation to recover cells that were irradiated in the S-G2. In each experiment,

CBs were evaluated for the following conditions: 1) control samples, 2) samples irradiated with priming dose $D_p$ alone, 3) samples irradiated with challenging dose $D_c$ alone and 4) samples irradiated with $D_p$ followed by $D_c$. The CBs distribution induced in neuroblasts following these four treatments was assessed by identifying chromatid deletions (CDs, scored as a single breaking event) and isochromosome breaks (ISOs, scored as two breaking events) (for examples of CDs and ISOs see Fig 2). The total number of CBs within a given cell was calculated as CBs = CDs + 2*ISOs.

The mean values of CBs per 100 cells along with their standard errors, were obtained from at least three independent experiments for both controls and samples pretreated with different priming doses and dose rates. Fig 3 shows statistical comparisons between pretreated samples and controls.

Our results showed that the mean values of CBs per 100 cells induced with priming doses up to 1.5 Gy at dose rates up to 17 mGy/h were not significantly different from that obtained for the control sample. These pretreatments did not produce a significant rise in CBs compared to the control value. However, pretreatment with a priming dose of 2.7 Gy at a dose rate of 17 mGy/h resulted in significantly higher CBs values compared to those for the control sample (Fig 3).

The mean values of CDs, ISOs and CBs per 100 cells together with their standard error observed for the different treatments at different $D_p$ followed by $D_c$ (from now on referred to as $D_p + D_c$) and for $D_c$ alone were obtained from at least three independent experiments. Fig 4 shows comparisons between the breaks obtained after $D_c$ without pretreatment and after each treatment ($D_p + D_c$).

The mean values of CBs per 100 cells for (Dp + Dc) were significantly lower than those for Dc alone at all priming dose rate values greater than 1.4 mGy/h, except for 0.4 Gy at 4.4 mGy/h where the differences were not significant. For both Dp of 0.1 Gy and 0.2 Gy at 1.4 mGy/h, no statistically significant differences were found in the mean values of ISOs and CBs per 100 cells induced by Dc with pretreatment and Dc alone. In contrast, the mean values of CDs per 100 cells induced by Dc with and without pretreatment do not align with the trends seen for ISOs and CBs. The variability of CD counts four hours after challenging irradiation may arise because CDs include both unrepaired CDs and ISOs where only one break was rejoined. This could explain the significantly higher CD values were observed for Dp of 0.1 Gy and 0.2 Gy at 1.4 mGy/h. During ISOs rejoining, a structure similar to simple CDs can form, which may be misclassified as residual CDs, thereby inflating CD counts [18].

### HRS/IRR experiments

In addition, for the first time, we studied the occurrence of CBs in larval neuroblasts induced by irradiating (non priming treated) third instar larvae with acute gamma rays at different doses in the range 0.7–10 Gy (dose rate of 0.65 Gy/min). CDs, ISOs and CBs were evaluated in the same way as for RAR experiments. The number of cells scored and the mean

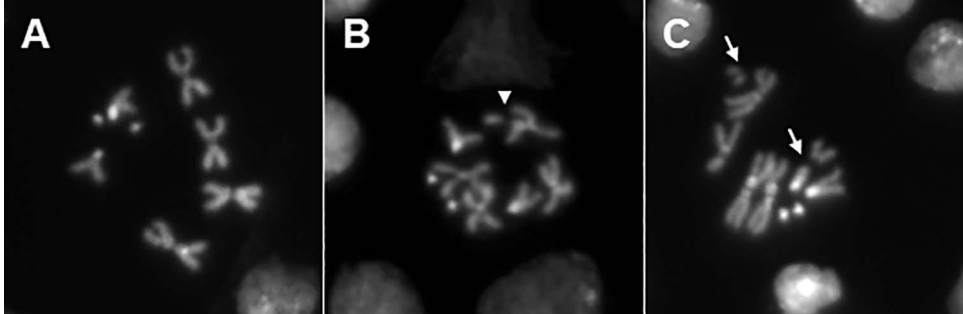

**Fig 2. Examples of chromosome breaks observed in neuroblasts from *Oregon* R third instar larvae.** (A) Wild-type female; (B) female metaphase showing autosomal chromatid deletions (arrow) (C) female metaphase with an isochromosome break affecting an autosome and another affecting the X chromosome (arrows).

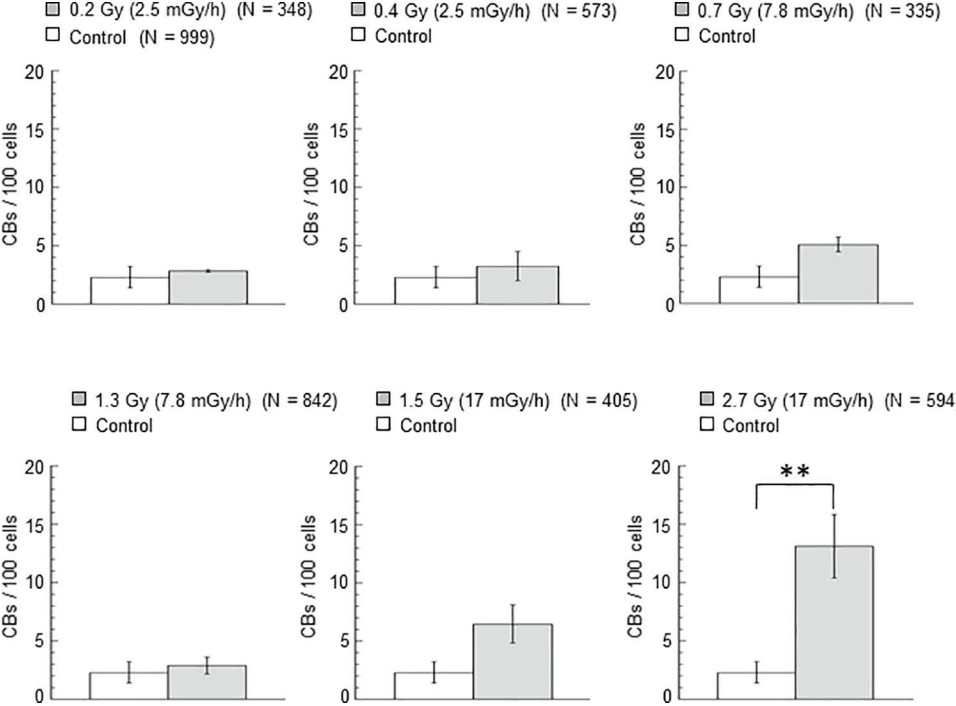

**Fig 3. Cytogenetic damage in the control and irradiated samples with different priming doses.** Analysis of the frequency of total CBs per 100 cells in larval neuroblasts from third instar larvae for control (non-irradiated) samples and for samples irradiated with different priming doses and priming dose rate but not subsequently irradiated with the challenging dose of 10 Gy. Mean values of CBs per 100 cells together with their standard errors, obtained from at least three independent experiments with at least 100 cells scored per experiment were reported. CDs and ISOs values have not been reported in the figure as they showed the same behaviour as CBs. N indicates the sum of the numbers of cells scored per treatment in the different experiments. Comparisons were performed between the values obtained in the control and the values obtained in each individual pre-treatment using Dunnett test after a one-way ANOVA test. Values of p < 0.01 (**) were considered as statistically significant.

values of CDs, ISOs and CBs per 100 cells were shown in Table 1. Chromosomes were fixed four hours post- acute irradiation to recover cells that were exposed during the S-G2 phase.

The dose response curve expressed as CBs frequency/100 cells vs dose is shown in Fig 5A.

To our knowledge, no previous study has reported such a dose-response curve. Surprisingly we observed hypersensitivity to radiation (HRS) at lower doses, followed by increased radioresistance (IRR) at higher doses; in fact, up to a dose of 2.7 Gy cells were more radiosensitive than predicted by back extrapolating high-dose response. However, at doses above 2.7 Gy, cells exhibited increased resistance (IRR).

To further highlight the low-dose HRS, we plotted the same data from Fig 5A in Fig 5B, expressing radiosensitivity per unit dose on the vertical axis.

The results clearly showed evidence of hypersensitivity for doses ≤ 2.7 Gy with peak HRS at 0.7 and 1.5 Gy. For these dose values, the radiosensitivity per unit dose of larval neuroblasts is approximately twice as high as the radiosensitivity per unit dose observed at dose ≥ 4 Gy.

## Discussion

In this article, we studied two typical phenomena that can occur when exposing a biological system to low doses/dose rates of ionizing radiation: RAR and HRS/IRR. An *in vivo* biological system given by the larvae of *Drosophila melanogaster* was considered. Our experimental design allowed us to examine the adaptive effect for low dose rate values of the

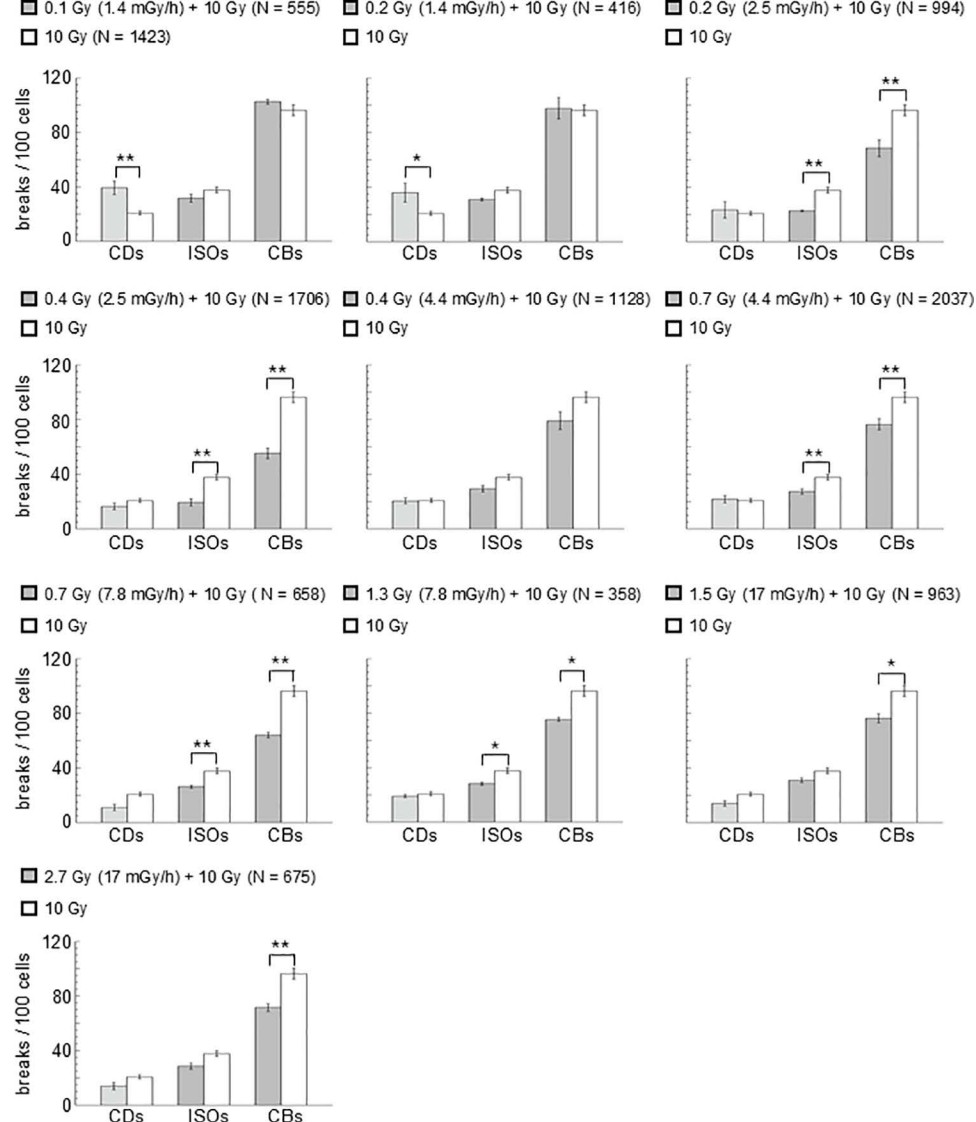

**Fig 4. Cytogenetic damage induced by challenging dose in pretreated and non-pretreated samples.** Comparison between mean values of breaks per 100 cells induced in larval neuroblasts by each individual treatment ($D_p + D_c$) and by $D_c$ alone. At least three independent experiments were performed for each condition with at least 100 cells scored per experiment. N indicates the total number of cells scored per treatment. The error bars represent the standard error of the mean. Dunnett test after a one-way ANOVA test was used to determine statistically significant differences. Values of $p < 0.05$ (*) and $p < 0.01$ (**) were considered as statistically significant.

priming dose. In particular, although we did not consider low priming dose values, we employed low priming dose rates (defined as <18 mGy/h). These values were less than or at most about threefold higher than the 5 mGy/h threshold used by UNSCEAR to define low dose rate provided for radiation such as external X-rays and gamma rays [19].

In our system, no RAR was observed at a priming dose rate of 1.4 mGy/h. Specifically, a priming dose of 0.2 Gy delivered at this rate did not induce adaptation, whereas this same dose at 2.5 mGy/h resulted in a clear RAR effect. RAR was found across priming doses ranging from 0.2 to 2.7 Gy and priming dose rates from 2.5 to 17 mGy/h, with the strongest adaptive effect at 0.4 Gy delivered at 2.5 mGy/h. These findings suggest the existence of threshold values for both

**Table 1. Analysis of the frequency of CBs in larval neuroblasts induced by irradiating third instar larvae with different doses D of acutely delivered gamma rays. Mean values of CDs, ISOs and CBs per 100 cells together with their standard errors, obtained from at least three independent experiments with at least 100 cells scored per experiment were reported.**

| D (Gy) | N | CDs/(100 cells) | ISOs/(100 cells) | CBs/(100 cells) |
|---|---|---|---|---|
| 0 | 1057 | 1.1±0.3 | 0.6±0.3 | 2.3±0.9 |
| 0.7 | 648 | 1.7±0.7 | 6.1±1.0 | 13.9±1.9 |
| 1.5 | 507 | 4.3±0.9 | 12.3±1.8 | 28.9±4.0 |
| 2.7 | 978 | 8.9±1.8 | 16.3±1.0 | 41.5±3.3 |
| 4.0 | 814 | 4.2±0.7 | 16.9±1.5 | 38.0±2.5 |
| 6.0 | 745 | 10.0±3.2 | 23.1±3.7 | 56.2±7.0 |
| 8.0 | 486 | 11.0±1.6 | 31.6±3.2 | 74.2±6.3 |
| 10.0 | 1423 | 20.8±1.5 | 37.8±2.2 | 96.4±3.9 |

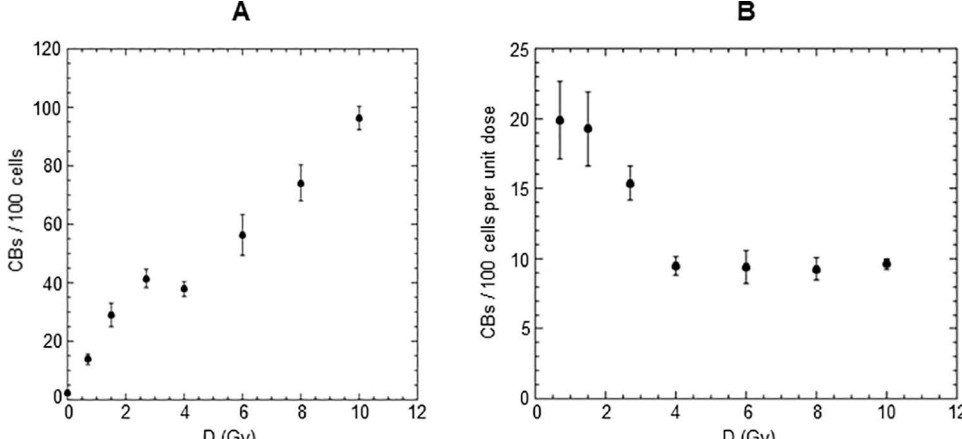

**Fig 5. Cytogenetic effect induced by different doses of gamma rays.** (A) Dose-response curves for CBs frequency/100 cells in larval neuroblasts from untreated third instar larvae irradiated with different doses of gamma rays (0.7, 1.5, 2.7, 4.0, 6.0, 8.0 and 10 Gy) at a high dose rate (0.65 Gy/min). Data points represent the means of at least three independent experiments with at least 100 cells scored per experiment. The error bars represent the standard error of the mean. (B) Radiation sensitivity of larval neuroblasts from untreated third instar larvae irradiated with different doses of gamma rays (0.7, 1.5, 2.7, 4.0, 6.0, 8.0 and 10 Gy). The data points for each dose D were obtained by dividing the average value of CBs/100 cells shown in (A) by the value of the dose D. The error bars represent the standard error of the mean.

priming dose (between 0.1 and 0.2 Gy) and dose rate (between 1.4 and 2.5 mGy/h) that must be exceeded for RAR to occur in our biological system.

Our results are consistent with previous *in vitro* and *in vivo* studies, which indicate that adaptive responses occur only when priming dose and dose rate fall within specific ranges [4]. Shadley and Wiencke observed that for a relatively high priming dose to be effective in inducing an adaptive response in human lymphocytes, it must be delivered at a rather low dose rate [20]. In *in vivo* studies, it was also found that the priming dose values resulting in an adaptive response seem to be located in a certain range. Liu et al. observed that whole-body irradiation of mice with low priming doses of X-rays in the range of 2–100 mGy at a dose rate of 57.3 mGy per minute induced an adaptive response in the bone marrow cells expressed as a reduction of chromosome aberrations following a second exposure to a larger dose. Moreover, they found that chronic whole-body gamma-irradiation of rabbits at a dose-rate of 92.6 µGy/min (5.6 mGy/h) induced a cytogenetic adaptive response in peripheral blood lymphocytes (for chromatid and isochromatid breaks) with an inductive dose of up to 1.5 Gy [21,22].

A possible explanation of our results could be based on the amount of oxidative stress induced per unit time on the biological system by the pretreatment. When the priming dose and the priming dose rate are too low, the levels of oxidative stress induced per unit time are too low and the cellular sensor of DNA damage fails to detect DSBs, leaving the DNA repair inactive and no RAR is observed. When the priming dose and priming dose rate values increase a level of oxidative stress induced per unit time is reached such that the cellular sensor of DNA damage can detect DSBs. This alerts the cell, giving a trigger signal to activate mechanisms that make the cells more radioresistant. Consequently, RAR can be observed. The ATM protein kinase represents a plausible candidate for this cellular sensor, as supported by experimental evidence and theoretical models [23,24]. However, this hypothesis requires experimental validation in our specific biological system.

For the first time, we studied the HRS/IRR phenomenon in *Drosophila melanogaster* larvae using chromosome breaks as endpoints. A dose-response curve revealed a distinct low-dose hypersensitivity (HRS) region at doses ≤2.7 Gy, followed by induced radioresistance (IRR) at higher doses.

HRS/IRR has been previously observed in *Drosophila* larvae [13], as well as in neutron-irradiated *Caenorhabditis elegans* [14] and mammalian tissues, including murine skin, kidney, and lung [25,26]. However, this phenomenon was observed mainly in *in vitro* studies, obtaining a significant reduction in clonogenic cell survival [9,10,27–30]. Only a few studies have linked HRS to increased chromosome breaks, micronuclei, and unrepaired DSBs [11,12,31]. Overall, for the *in vitro* studies, the HRS response was typically obtained at doses lower than about 0.4 Gy when given at acute dose rates. In particular, for chromosome breaks in *in vitro* biological systems, 0.4 Gy was the highest dose for which HRS was observed, whereas for our *in vivo* system such HRS response was obtained up to a dose of 2.7 Gy. This difference could be related to the high intrinsic radioresistance of *Drosophila* larvae. Although the dose values, at which the HRS phenomenon occurs, were different for our *in vivo* system and for other *in vitro* systems, the underlying mechanisms of HRS/IRR are likely similar across them. Consistent with *in vitro* findings, we observed HRS/IRR in neuroblasts irradiated during the G2 phase, a stage previously identified as particularly sensitive to this phenomenon [32]. Both RAR and HRS/IRR in *Drosophila* larvae could be determined by the same mechanism related to DNA repair efficiency, as hypothesized by in *in vitro* studies [29,33]. Further studies will be needed to clarify these relationships.

Finally, the damage in terms of chromosome breaks in larval neuroblasts from third instar larvae obtained at the 2.7 Gy dose delivered acutely (about 45 CBs/100 cells) was much greater than that observed by delivering the same 2.7 Gy dose chronically from embryos until the formation of third instar larvae (about 13 CBs/100 cells). This aligns with several studies with *in vivo* and *in vitro* biological systems that predict, at the same dose, a reduction in damage for low dose rate exposures compared to high dose rate exposures [34–37]. However, it contradicts other studies that predict an inverse dose rate effect [38–40]. This could be due to several reasons, e.g., different biological systems, experimental setup, end points, etc.

In conclusion, these results could provide important information about the mechanisms underlying HRS and RAR in *Drosophila*. This *in vivo* system proved to be very suitable for studying these phenomena at low doses/dose rates (e.g., using genomics and transcriptomics techniques). Such research could have broad implications for all situations involving prolonged exposure to radiation such as in earth and space radiation protection, nuclear medicine diagnostics and theragnostics, and biodosimetry. For instance, it could offer potential strategies for mitigating the risks of long-duration space missions, where astronauts are continuously exposed to low doses of ionizing radiation but may also incur high-dose exposure in case of solar events [41]. During space travel, prolonged low-dose radiation exposure may induce cellular adaptation, potentially increasing resistance to subsequent higher doses [42,43]. In contrast, the phenomenon of HRS/IRR could theoretically amplify the risks posed by minor solar events [44]. Beyond spaceflight, studies on radioadaptive response can potentially lead to new treatments or strategies to mitigate radiation damage in diagnostics and theragnostics [45]. Moreover, the study of phenomena such as RAR and HRS/IRR may help reduce errors in biodosimetry by conventional cytogenetics [46].

## Supporting information

**S1 Fig. Design of RAR experiment.** Vials containing Drosophila embryos were placed in the LIBIS irradiator at different distances from the 137Cs source and exposed to continuous irradiation at different dose rates until they developed into third instar larvae. 89 and 161 hours after the start of the priming exposure, vials (containing third instar larvae) were taken out of LIBIS and some of them were irradiated with an additional 10 Gy of γ-rays (challenging dose). At both the time points, even unpretreated vials containing third instar larvae were exposed to the challenging dose alone. Moreover, vials that were not exposed to either the priming dose or the challenging dose were also considered in our study. Subsequently, all vials were analysed to determine the frequency of CBs.
(TIF)

**Raw data. Dataset to fully reproduce the results of this study.**
(XLSX)

## Acknowledgments

The authors are indebted to P. Anello (Istituto Superiore di Sanità) for technical assistance.

## Author contributions

**Conceptualization:** Giuseppe Esposito, Claudia Di Dio, Antonella Porrazzo, Maria Antonella Tabocchini, Giovanni Cenci, Francesca Cipressa.

**Data curation:** Claudia Di Dio, Antonella Porrazzo, Alex De Gregorio.

**Formal analysis:** Giuseppe Esposito, Claudia Di Dio.

**Investigation:** Claudia Di Dio, Antonella Porrazzo, Alex De Gregorio, Patrizia Morciano.

**Methodology:** Giuseppe Esposito, Claudia Di Dio, Antonella Porrazzo, Giovanni Cenci, Francesca Cipressa.

**Supervision:** Giuseppe Esposito, Giovanni Cenci, Francesca Cipressa.

**Writing – original draft:** Giuseppe Esposito.

**Writing – review & editing:** Giuseppe Esposito, Claudia Di Dio, Antonella Porrazzo, Patrizia Morciano, Maria Antonella Tabocchini, Giovanni Cenci, Francesca Cipressa.

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
