## [Decision Letter · Decision Letter 0]

30 Apr 2025

PONE-D-25-11866*In vivo* study of the radioadaptive response and low-dose hyper-radiosensitivity for chromosome breaks induced by gamma rays in wild-type *Drosophila melanogaster* larval neuroblasts: dose and dose rate dependencePLOS ONE

Dear Dr. esposito,

Thank you for submitting your manuscript to PLOS ONE. After careful consideration, we feel that it has merit but does not fully meet PLOS ONE’s publication criteria as it currently stands. Therefore, we invite you to submit a revised version of the manuscript that addresses the points raised during the review process.

We look forward to receiving your revised manuscript.

Kind regards,

Amit Singh, PhD

Academic Editor

PLOS ONE

Journal Requirements:

3. Thank you for stating the following financial disclosure: [The author(s) declare financial support was received for the research, authorship, and/or publication of this article. This study was supported by Istituto Superiore di Sanità (Italian National Institutes of Health)]. 

4. We note that your Data Availability Statement is currently as follows: [All relevant data are within the manuscript and its Supporting Information files]

Additional Editor Comments:

As you can see there is significant interest in your paper please address the concerns of the reviewers.

Reviewers' comments:

Reviewer's Responses to Questions

**Comments to the Author**

1. Is the manuscript technically sound, and do the data support the conclusions?

Reviewer #1: Yes

Reviewer #2: Yes

2. Has the statistical analysis been performed appropriately and rigorously? 

Reviewer #1: No

Reviewer #2: Yes

3. Have the authors made all data underlying the findings in their manuscript fully available?

Reviewer #1: Yes

Reviewer #2: Yes

4. Is the manuscript presented in an intelligible fashion and written in standard English?

Reviewer #1: Yes

Reviewer #2: Yes

5. Review Comments to the Author

Reviewer #1: PONE-D-25-11866

MANUSCRIPT SUMMARY

This manuscript is a study on a rather specific area of LDR exposure, primarily aimed at establishing a threshold dose rate at which radioadaptive responses can be measured using the D. melanogaster model. This group have published a rather elegant study on this topic previously, the current study is not quite as impactful. However, the authors explicitly state on several occasions what new data is in this manuscript and how it may be useful to the rad field at large.

On this point, the authors state in the abstract:

“Our results provide insight into the complex cellular responses to low-dose/dose rate

radiation and have implications in various fields, including radiation protection, diagnostics, theragnostics and biodosimetry”

Rather than making this rather broad statement, it would be useful to go through each one of these topics and give a specific application of the current work to that subfield in radiation and provide references. This could be done in either the introduction or the discussion. By adding some relevant discussion on the application of this study it will be more accessible to non-specialists.

Specific Comments:

I appreciate Figure 1. But the remainder of the study design can still be a bit difficult to follow. Another timeline figure should be included in the supplement to outline the study design.

There are sometimes disagreements on the real biological effects between using X ray or gamma sources. I appreciate that this is considered in the discussion and it is a useful comparison to have.

I can’t tell exactly how many t tests were conducted, but shouldn’t a corrected p value be used rather than 0.05 for multiple t tests?

Reviewer #2: In the manuscript entitled “In vivo study of the radioadaptive response and low-dose hyper-radiosensitivity for chromosome breaks induced by gamma rays in wild-type Drosophila melanogaster larval neuroblasts: dose and dose rate dependence”, the authors aim to investigate the phenomenon of radioadaptive response (RAR) in Drosophila melanogaster larval neuroblasts, focusing on the influence of different gamma priming doses and priming dose rates. Additionally, the authors investigated hyper-radiosensitivity and increased radioresistance by performing in vivo experiments. This article provides novel information to the scientific community. In particular, highlighting the novel use of Drosophila melanogaster for HRS/IRR analysis using chromosome breaks as an endpoint is commendable and potentially impactful. However, the manuscript is ambiguous, and there are some points the authors should carefully consider before publication.

Methods

Irradiation treatments:

• The manuscript will benefit from a clear rationale for the selection of specific doses and dose rates used in both the protracted and acute irradiation protocols.

• Redundant information: The incubation procedure is repeated verbatim.

• Line 108: Possibly typographical error; the authors probably mean 1.5 and not 1,5?

Drosophila chromosome cytology and microscopy

• What were the conditions in which dissected Drosophila larval brains were treated for 1 h with 10-6 M colchicine?

• Were the samples fixed before or after the hypotonic treatment?

Discussion

• The authors should clearly define what constitutes low dose and low dose-rate. In the Introduction (lines 36–37), they cite Wilson et al. and suggest thresholds of <100 mGy for low dose and <5 mGy/h for low dose-rate; however, this definition should be explicitly stated and justified within the context of their study.

• Line 200: “Our experimental design allowed us to examine the adaptive effect for very low dose rate values of the priming dose.” What constitutes VERY low dose-rate values? Lowest used in the work is 1.4 mGy/h.

• Lines 219-227: The mechanistic interpretation, while plausible, appears somewhat generalized and would benefit from stronger support through relevant literature or experimental data. In the absence of direct molecular evidence, the explanation remains largely speculative.

• The “Introduction” and “Discussion” looked somewhat tedious; please simplify them.

• The reviewer suggests reducing redundancies for clarity.

6. PLOS authors have the option to publish the peer review history of their article (what does this mean? ). If published, this will include your full peer review and any attached files.

**Do you want your identity to be public for this peer review?** For information about this choice, including consent withdrawal, please see our Privacy Policy .

Reviewer #1: No

Reviewer #2: **Yes: ** Naresh Deoli

---

## [Author Response · Author response to Decision Letter 1]

14 May 2025

Reviewer #1: PONE-D-25-11866

MANUSCRIPT SUMMARY

This manuscript is a study on a rather specific area of LDR exposure, primarily aimed at establishing a threshold dose rate at which radioadaptive responses can be measured using the D. melanogaster model. This group have published a rather elegant study on this topic previously, the current study is not quite as impactful. However, the authors explicitly state on several occasions what new data is in this manuscript and how it may be useful to the rad field at large.

We are grateful to this reviewer for his/her potential appreciation of our work.

On this point, the authors state in the abstract:

“Our results provide insight into the complex cellular responses to low-dose/dose rate

radiation and have implications in various fields, including radiation protection, diagnostics, theragnostics and biodosimetry”

Rather than making this rather broad statement, it would be useful to go through each one of these topics and give a specific application of the current work to that subfield in radiation and provide references. This could be done in either the introduction or the discussion. By adding some relevant discussion on the application of this study it will be more accessible to non-specialists.

We followed this reviewer's suggestion by adding the following comment at the end of the discussion:

Such research could have broad implications for all situations involving prolonged exposure to radiation such as in earth and space radiation protection, nuclear medicine diagnostics and theragnostics, and biodosimetry. For instance, it could offer potential strategies for mitigating the risks of long-duration space missions, where astronauts are continuously exposed to low doses of ionizing radiation but may also incur high-dose exposure in case of solar events [41]. During space travel, prolonged low-dose radiation exposure may induce cellular adaptation, potentially increasing resistance to subsequent higher doses [42, 43]. In contrast, the phenomenon of HRS/IRR could theoretically amplify the risks posed by minor solar events [44]. Beyond spaceflight, studies on radioadaptive response can potentially lead to new treatments or strategies to mitigate radiation damage in diagnostics and theragnostics [45]. Moreover, the study of phenomena such as RAR and HRS/IRR may help reduce errors in biodosimetry by conventional cytogenetics [46].

Specific Comments:

I appreciate Figure 1. But the remainder of the study design can still be a bit difficult to follow. Another timeline figure should be included in the supplement to outline the study design.

Following the reviewer's suggestion, we have included a figure in the supplement (Fig S1)

There are sometimes disagreements on the real biological effects between using X ray or gamma sources. I appreciate that this is considered in the discussion and it is a useful comparison to have.

We thank the reviewer for the comment

I can’t tell exactly how many t tests were conducted, but shouldn’t a corrected p value be used rather than 0.05 for multiple t tests?

We thank the reviewer for pointing this out. In our study we want to compare each treatment group only with a single group. For figure 3, we compared each individual treatment given by the samples exposed to the different priming doses, with the control treatment given by the non-irradiated samples. For figure 4, we compared each individual treatment given by the samples exposed to the different priming doses followed by the challenging dose, with the single treatment given by the samples exposed to the challenging dose alone. We agree that for our multiple t-tests, a corrected p-value should be used instead of 0.05. Consequently, we considered replacing t-tests with one-way ANOVA tests followed by Dunnett's tests to reduce the possibility of type I error (false positives).

We have modified the text of section “Statistical Analysis” and figures 2 and 3 accordingly as follows:

Statistical Analysis

…The average of the mean values of CBs per cell obtained from at least three independent experiments was calculated together with its standard error (SE) for each condition. To determine statistical significant differences between pairs of these averages, the Graphpad software was used by performing a Dunnett test after a one-way ANOVA test. a parametric test (unpaired Student’s t-test) was used. Values of…

Fig 3. …..”Comparisons were performed between the values obtained in the non-irradiated control and the values obtained in each individual pre-treatment using Student’s t-test Dunnett test after a one-way ANOVA test. Values of p < 0.05 (*) p < 0.01 (**) were considered as statistically significant”.

Fig 4…… “Comparison between mean values of breaks CDs, ISOs and total CBs per 100 cells induced in larval neuroblasts by each individual treatment (Dp + Dc) and by Dc alone the acute exposure to 10 Gy in chronically pretreated and non-pretreated third instar larvae. At least three independent experiments were performed for each condition with at least 100 cells scored per experiment. N indicates the total number of cells scored per treatment. The error bars represent the standard error of the mean. Dunnett test after a one-way ANOVA test was used to determine statistically significant differences. Values of p < 0.05 (*), and p < 0.01 (**) and p < 0.001 (***) were considered as statistically significant”.

Moreover we have corrected figures 3 and 4 according to the output of the ANOVA test followed by the Dunnet test.

Finally, we added the following comment at the end of the ‘Results’ section:

The mean values of CBs per 100 cells for (Dp+Dc) were significantly lower than those for Dc alone at all priming dose rate values greater than 1.4 mGy/h, except for 0.4 Gy at 4.4 mGy/h where the differences were not significant. For both Dp of 0.1 Gy and 0.2 Gy at 1.4 mGy/h, no statistically significant differences were found in the mean values of ISOs and CBs per 100 cells induced by Dc with pretreatment and Dc alone. In contrast, the mean values of CDs per 100 cells induced by Dc with and without pretreatment do not align with the trends seen for ISOs and CBs. The variability of CD counts four hours after challenging irradiation may arise because CDs include both unrepaired CDs and ISOs where only one break was rejoined. This could explain the significantly higher CD values were observed for Dp of 0.1 Gy and 0.2 Gy at 1.4 mGy/h. During ISOs rejoining, a structure similar to simple CDs can form, which may be misclassified as residual CDs, thereby inflating CD counts [18].

Reviewer #2:

In the manuscript entitled “In vivo study of the radioadaptive response and low-dose hyper-radiosensitivity for chromosome breaks induced by gamma rays in wild-type Drosophila melanogaster larval neuroblasts: dose and dose rate dependence”, the authors aim to investigate the phenomenon of radioadaptive response (RAR) in Drosophila melanogaster larval neuroblasts, focusing on the influence of different gamma priming doses and priming dose rates. Additionally, the authors investigated hyper-radiosensitivity and increased radioresistance by performing in vivo experiments. This article provides novel information to the scientific community. In particular, highlighting the novel use of Drosophila melanogaster for HRS/IRR analysis using chromosome breaks as an endpoint is commendable and potentially impactful. However, the manuscript is ambiguous, and there are some points the authors should carefully consider before publication.

We are grateful to this reviewer for his/her potential appreciation of our work and for his/her suggestions that are certainly useful to improve the manuscript. We have revised the manuscript accordingly.

Methods

Irradiation treatments:

• The manuscript will benefit from a clear rationale for the selection of specific doses and dose rates used in both the protracted and acute irradiation protocols.

As suggested by the reviewer, we have added a rationale for the selection of doses and dose rates used in the experiments in the section “Irradiation treatments”:

….We have also collected third instar larvae coming out after 161 h of maintenance inside LIBIS at d1, d2, d3, d4 and d5 distances that resulted into a total priming dose of 0.2 Gy (at 1.4 mGy/h), 0.4 Gy (at 2.5 mGy/h), 0.7 Gy (at 4.4 mGy/h), 1.3 Gy (at 7.8 mGy/h) and 2.7 Gy (at 17 mGy/h), respectively. The selected dose (0.4 Gy) and dose rate (2.5 mGy/h) values were based on previous findings demonstrating clear radioadaptive response [8]. This experimental design allows us to investigate whether our biological system exhibits adaptation within a specific range of priming dose and dose rate values….

… to an acute challenging gamma radiation dose of 10 Gy at a dose rate of 0.65 Gy/min (values already used in the previous study [8]). Following….

• Redundant information: The incubation procedure is repeated verbatim.

We have corrected as follows:

… at a dose rate of 0.65 Gy/min. Also in this case, following acute irradiation, the same procedure as described above was followed. non-irradiated and irradiated larvae were maintained in the same incubator at 25 °C for four hours and then they were dissected for the analysis of CBs frequency in neuroblasts. Only female larvae were considered in the analysis.

• Line 108: Possibly typographical error; the authors probably mean 1.5 and not 1,5?

We have corrected the typo

Drosophila chromosome cytology and microscopy

• What were the conditions in which dissected Drosophila larval brains were treated for 1 h with 10-6 M colchicine?

We agree with the reviewer about providing more methodological information in this section and added more details

To obtain metaphase chromosome preparations from Drosophila larval neuroblasts for the analysis of chromosome aberrations in metaphase mitotic chromosomes, third instar larval brains were dissected in a drop of 0.7% NaCl 0.7%. The isolated brains were then transferred to a Petri dish containing and treated for 1 h with 10−6 M colchicine. Samples were incubated for 1 hour at room temperature. …

• Were the samples fixed before or after the hypotonic treatment?

…Following incubation, brains were fixed Successively, they were transferred in 0.5% sodium citrate a hypotonic solution (0.5% sodium citrate) for 5 minutes and then they were transferred to a small drop of squashed in 45% acetic acid on coverslip and immediately frozen in liquid nitrogen….

Discussion

• The authors should clearly define what constitutes low dose and low dose-rate. In the Introduction (lines 36–37), they cite Wilson et al. and suggest thresholds of <100 mGy for low dose and <5 mGy/h for low dose-rate; however, this definition should be explicitly stated and justified within the context of their study.

In our study we consider low values of priming dose rate defined as less than 18 mGy/h. These values are less than or at most about three times higher than the value of 5 mGy/h corresponding to the definition of low dose rates given by UNSCEAR for radiations such as external X-rays and gamma rays. The cumulative priming dose values used are greater than 100 mGy so we consider them as intermediate or high doses. We have added a comment to clarify that point:

…low dose rate values of the priming dose. In particular, although we did not consider low priming dose values, we employed low priming dose rates (defined as <18 mGy/h). These values were less than or at most about threefold higher than the 5 mGy/h threshold used by UNSCEAR to define low dose rate provided for radiation such as external X-rays and gamma rays [19]. In our system, no RAR was…

• Line 200: “Our experimental design allowed us to examine the adaptive effect for very low dose rate values of the priming dose.” What constitutes VERY low dose-rate values? Lowest used in the work is 1.4 mGy/h.

We agree with this reviewer comment about the over statement of VERY low dose rate. We have deleted the term “very”

• Lines 219-227: The mechanistic interpretation, while plausible, appears somewhat generalized and would benefit from stronger support through relevant literature or experimental data. In the absence of direct molecular evidence, the explanation remains largely speculative.

Following the reviewer's suggestion, we have simplified this part of the discussion and inserted literature data as follows:

…A possible explanation of our results could be based on the amount of damage oxidative stress induced per unit time on the biological system by the pretreatment, which increases as the priming dose and priming dose rate values increase. There may be appropriate threshold values of the priming dose, Dp(s), and the priming dose rate, dD/dtp (s). When the priming dose and the priming dose rate are too low below Dp(s) and dD/dtp (s), the levels of oxidative stress DNA damage induced per unit time are too low and the cellular sensors of DNA damage fails to detect DSBs them, leaving the DNA repair inactive and no RAR is observed. When the priming dose and priming dose rate values increase and become greater than Dp(s) and dD/dtp (s) a level of oxidative stress damage induced per unit time is reached such that which the cellular sensors of DNA damage can detect DSBs. This alerts the cell, giving a trigger signal to activate mechanisms that make the cells more radioresistant, for example triggering faster or more efficient DNA repair. Consequently, RAR can be observed. The ATM protein kinase represents a plausible candidate for this cellular sensor, as supported by experimental evidence and theoretical models [23, 24]. However, this hypothesis requires experimental validation in our specific biological system. Evidence of the involvement of a more efficient DSB repair system in this radioadaptive response phenomenon has already been provided previously for the priming dose of 0.4 Gy delivered at 2.5 mGy/h [8]. Future research should aim to identify the specific DNA damage sensors and pathways responsible for triggering this process. For the first time…

• The “Introduction” and “Discussion” looked somewhat tedious; please simplify them.

Following the reviewer's suggestion, we have simplified the sections “Introduction” and “Discussion”.

• The reviewer suggests reducing redundancies for clarity.

Following the reviewer's suggestion, we reduced redundancies.

---

## [Editor Report · Decision Letter 1]

15 May 2025

*In vivo* study of the radioadaptive response and low-dose hyper-radiosensitivity for chromosome breaks induced by gamma rays in wild-type *Drosophila melanogaster* larval neuroblasts: dose and dose rate dependence

PONE-D-25-11866R1

Dear Dr. esposito,

We’re pleased to inform you that your manuscript has been judged scientifically suitable for publication and will be formally accepted for publication once it meets all outstanding technical requirements.

Kind regards,

Amit Singh, PhD

Academic Editor

PLOS ONE
---

## [Editor Report · Acceptance letter]

PONE-D-25-11866R1

PLOS ONE

Dear Dr. Esposito,

I'm pleased to inform you that your manuscript has been deemed suitable for publication in PLOS ONE. Congratulations! Your manuscript is now being handed over to our production team.

Kind regards,

on behalf of

Dr. Amit Singh

Academic Editor

PLOS ONE